# MHP-DDP: Multivariate Hawkes Process based on Dependent Dirichlet Process

**Alex Ziyu Jiang**
University of Washington
jiang14@uw.edu

**Abel Rodriguez**
University of Washington
abelrod@uw.edu

## Abstract

Multivariate Hawkes Processes (MHPs) model complex temporal dynamics among event sequences on multiple dimensions. Typically, strong parametric assumptions are made about the excitation functions of MHP, motivating the need for modeling flexible excitation patterns. Further, different excitation functions across dimensions often have strong similarities. Motivated by reasons above, we propose *MHP based on dependent Dirichlet process* (MHP-DDP), a hierarchical nonparametric Bayesian modeling approach for MHP. MHP-DDP flexibly estimates the excitation function via a mixture of scaled Beta distributions, and borrows strengths across dimensions by modeling such mixing distribution as a mixture of a shared Dirichlet process (DP) and a group-specific idiosyncratic DP. We develop two algorithms using Markov chain Monte Carlo (MCMC) and the stochastic variational inference (SVI) algorithm. We also conduct simulations to compare MHP-DDP to benchmark methods where total or no information is borrowed. We show that MHP-DDP outperforms the benchmark methods in terms of lower estimation error for both algorithms, with SVI being computationally efficient than MCMC.

## 1 Introduction

Point processes (e.g., see Cox and Isham, 1980) are a widely used to model temporal event sequence data, i.e., data corresponding to the times of occurrence of a series of countable events. In particular, multivariate Hawkes Processes (MHPs, e.g., see Hawkes, 1971 and Liniger, 2009) have been widely used to model sequences of events that demonstrate *self-* and *mutually-exciting* behaviors, i.e., patterns in which the likelihood of events increase after the occurrence of others. Hawkes Processes have been widely applied in a wide range of fields, including seismology (Ogata, 1988), finance (Bacry et al., 2015), electronic health records (Choi et al., 2015; Sun et al., 2024) and social media analysis (Rizoiu et al., 2017). MHPs can be characterized through their conditional intensity functions, which describe the instantaneous rate of arrivals of new events. The excitation function is an important module of the conditional intensity function, as it controls how past events cause the conditional intensity function to change and decay. Excitation functions are often modeled parametrically using exponential functions that assume monotonic excitation decay (Hawkes, 1971). However, the inter-event times in many real-world examples tend to have complex patterns, motivating the need for flexible excitation functions (e.g., see Markwick, 2020 and Rodríguez et al., 2017).

In this paper, we introduce a Bayesian nonparametric model for MHPs that builds on the ideas of Donnet et al. (2020) and Markwick (2020) but addresses various practical questions that have so far remained open. One challenge associated with the estimation of MHPs is that the number of parameters that need to be estimated grows quadratically with the number of dimensions. Hence, with a

Workshop on Bayesian Decision-making and Uncertainty, 38th Conference on Neural Information Processing Systems (NeurIPS 2024).

dataset of moderate size, modeling each dimension of the process independently can lead to ineffi-
ciencies. Motivated by the observation that often excitation functions looks similar across different
dimensions, we propose a hierarchical modeling approach based on mixtures of non-parametric mix-
tures. More specifically, we adapt the approach introduced in Müller et al. (2004), which models
each of the excitation functions as a mixture of an idiosyncratic component and a common compo-
nent shared by all excitation functions, and study some of the properties of such formulation.

A second challenge in implementing MHP models is computation. As is the case more generally,
Markov chain Monte Carlo (MCMC) algorithms are the most common approach to computation for
Bayesian models for Hawkes processes (e.g., see Rasmussen, 2013). However, MCMC algorithms
for Hawkes process models are often to be too slow even for moderate sample sizes because, except
for special cases such as the exponential excitation function, their complexity is quadratic in both
the number of observations and the number of dimensions. This challenge is amplified in the case of
nonparametric models. To address it, we expand on previous work on the use of stochastic gradient
methods for MHPs, and develop a scalable stochastic variational inference (SVI) algorithm that can
be used to fit our model. An important part of this development involves a carefully comparison of
the performace of SVI and MCMC methods, both in terms of accuracy and speed.

In summary we make two key contributions in this paper: (1) we propose a novel and flexible model
for linear MHPs in which the various excitation functions are assigned a joint nomparametric prior
that allows us to efficiently borrow information, (2) we develop MCMC and SVI algorithms for
estimation and prediction in the context of this nonparametric model, and thoroughly evaluate their
relative performance.

## 2  Multivariate Hawkes Processes

Let $N^{(1)}(t), \ldots, N^{(K)}(t)$ be a collection of $K$ point processes defined on the positive real line $\mathbb{R}^+$,
where $N^{(k)}(t)$ represents the number of events on dimension $k$ to occur on the interval $[0, t]$. We
denote a generic set of observations from this process by $\mathbf{X} = \{(t_i, d_i) : i = 1, \ldots, n\}$, where
$t_i \in \mathcal{R}^+$ represents the timestamp at which the $i$-th event occurs and $d_i \in \{1, \ldots, K\}$ represents
the dimension in which the event occurs. Then, $\mathbf{X}$ follows a multivariate Hawkes process (Hawkes,
1971; Liniger, 2009) if the conditional intensity function on dimension $k$ has the following form:

$$\lambda_k(t) \equiv \lim_{h \to 0} \frac{\mathbb{E}\left[N^{(k)}(t+h) - N^{(k)}(t) \mid \mathcal{H}_t\right]}{h} = \mu_k + \sum_{k=1}^{K} \sum_{t_i < t, d_i = k} \alpha_{\ell,k} \tilde{\phi}_{\ell,k}(t - t_i),$$

where we $\mu_k > 0$ is the background intensity for dimension $k$, $\alpha_{\ell,k} > 0$ be the parameter that
controls the strength by which past events from dimension $\ell$ influence the occurrence of new events
on dimension $k$, and $\tilde{\phi}_{\ell,k}(\cdot) : \mathbf{R}^+ \to \mathbf{R}^+$ be the (normalized) excitation function that controls how
such influence decays over time. Note that we require that $\int_0^\infty \tilde{\phi}_{\ell,k}(s)\mathrm{d}s = 1$, which ensures that
$\tilde{\phi}_{\ell,k}$ are identifiable. An alternative construction of the MHP is as a multivariate branching process
in which the first generation of events in dimension $k$ (often called 'immigrants" in the literature)
arise from a homogeneous Poisson process with rate $\mu_k$, and the points in subsequent generations are
generated from non-homogenous Poisson processes with rates given by the $\alpha_{\ell,k}$s and the interarrival
times are controlled by the $\phi_{\ell,k}$s. In the sequel, we use the binary matrix $\mathbf{B}$ where

$$B_{j,j} = \begin{cases} 1 & j\text{-th event is an immigrant} \\ 0 & \text{otherwise} \end{cases}, B_{i,j} = \begin{cases} 1 & j\text{-th event is generated from } i\text{-th event} \\ 0 & \text{otherwise} \end{cases}$$

to encode the latent branching structure associated with a realization of an MHP. The augmented
likelihood for the data is then given by

$$\mathcal{L}(\mathbf{X}, \mathbf{B} \mid \boldsymbol{\alpha}, \boldsymbol{\phi}) = \sum_{k=1}^{K} \sum_{\ell=1}^{K} \left[ |O_{k,\ell}| \left(\log \alpha_{k,\ell}\right) - \sum_{\substack{i<j \\ d_i=k, d_j=\ell}} B_{i,j} \tilde{\phi}_{k,\ell}(t_j - t_i) \right]$$

$$+ \sum_{\ell=1}^{K} |I_\ell| \log \mu_\ell - \sum_{\ell=1}^{K} \mu_\ell T - \sum_{k=1}^{K} \sum_{\ell=1}^{K} \alpha_{k,\ell} \sum_{i:d_i=k} \tilde{\Phi}_{k,\ell}(T - t_i), \quad (1)$$

where $|I_\ell| = \sum_{d_i = \ell} I(B_{ii} = 1)$ and $|O_{k,\ell}| = \sum_{d_i = k, d_j = \ell, i < j} I(B_{ij} = 1)$ denote the number of immigrants for dimension $k$ and the number of offpring on dimension $k$ who arise from points on dimension $\ell$.

# 3 Nonparametric Bayesian modeling of excitation functions for multivariate Hawkes processes

First, we note that for all $k, \ell = 1, \ldots, K$, $\tilde{\phi}_{k,\ell}$ can be interpreted as a probability distribution function on $\mathbb{R}^+$, with corresponding cumulative distribution function $\tilde{\Phi}_{k,\ell}$. Denote $\mathcal{H}$ as the space of all density functions on $\mathbb{R}$, MHP-DDP defines a prior on $\tilde{\phi}_{k,\ell}$ over $\mathcal{H}$ as a mixture of scaled Beta distributions, with respect to a random measure $\mathrm{G}_{k,\ell}(\cdot)$:

$$\tilde{\phi}_{k,\ell}(t) = \int_{\mathbb{R}^2} f_{\text{Beta}}(t \mid a, b, T_0) d\mathrm{G}_{k,\ell}(a, b), \tag{2}$$

where the kernel density function

$$f_{\text{Beta}}(\cdot \mid a, b, T_0) := \frac{\Gamma(a+b)}{\Gamma(a)\Gamma(b)} \left(\frac{t}{T_0}\right)^{a-1} \left(1 - \frac{t}{T_0}\right)^{b-1} T_0^{-1}, \ \ a > 0, b > 0, 0 < t < T_0$$

is a scaled Beta density function with support on $(0, T_0)$, indexed by shape parameters $a$ and $b$. The Beta mixture allows the excitation function to have a flexible form, including multi-modal structures and heavy tails.

We let $\mathrm{G}_{k,\ell}(\cdot)$ be the mixing distribution for the two shape parameters that corresponds to $\tilde{\phi}_{k,\ell}$, which is a distribution on $\mathbb{R}^2$. Denote $\mathcal{M}(\mathbb{R}^2)$ as the space of all probability distributions on $\mathbb{R}^2$, we define a prior on $\mathrm{G}_{k,\ell}$ over $\mathcal{M}(\mathbb{R}^2)$ using the following hierarchical model, following Müller et al. (2004):

$$\begin{aligned}
\mathrm{G}_{k,\ell}(\cdot) &= \varepsilon \mathrm{H}_0(\cdot) + (1 - \varepsilon)\mathrm{H}_{k,\ell}(\cdot), \ \ k, \ell = 1, \ldots, K, \\
\mathrm{H}_0, \mathrm{H}_{k,\ell} &\sim \mathrm{DP}(\alpha_{\mathrm{DP}}, \mathrm{Gamma}\,(c_a, d_a) \times \mathrm{Gamma}\,(c_b, d_b)) \ \ k, \ell = 1, \ldots, K, \\
\varepsilon &\sim \mathrm{Beta}(1, 1).
\end{aligned} \tag{3}$$

Under the hierarchical prior, $\mathrm{G}_{k,\ell}$ can be expressed as a mixture of two random measures: the common component $H_0$ that is shared across all dimension pairs and the idiosyncratic component $H_{k,\ell}$ that characterizes the dimension-pair specific behaviors of $\tilde{\phi}_{k,\ell}$. We let $\varepsilon \in [0, 1]$ be the weighting factor that controls how much prior information on the characteristics of the distributions is borrowed across all dimension pairs, and assume it follows a uniform prior between 0 and 1. Finally, we let $H_0$ and all $H_{k,\ell}, k, \ell = 1, \ldots, K$ be independent random measures drawn from the Dirichlet process prior with concentration parameter $\alpha_{\mathrm{DP}}$ and base measure $\mathrm{Gamma}\,(c_a, d_a) \times \mathrm{Gamma}\,(c_b, d_b)$, where $c_a, d_a, c_b, d_b$ are fixed hyperparameters that correspond to the two shape parameters of the Beta distribution. We note that there are two extreme cases of special interest: If $\varepsilon = 1$, all dimension pairs share the same triggering kernel, and when $\varepsilon = 0$, $\mathrm{G}_{k,\ell}, k, \ell = 1, \ldots, K$ are independently drawn from the same DP prior and share information only through the common hyperparameters in the base measure. Finally, we showed that the prior on the dependent Dirichlet mixture of scaled Beta kernels in (2) and (3) satisfies the Kulback-Leibler property model. The theorem and its proof is outlined in Appendix A.

For implementation, we consider a finite number truncation approximation on the number of mixtures (Ishwaran and James, 2002) for the DP mixture considered in (2). Further, we assign Gamma priors to the Hawkes process parameters:

$$\begin{aligned}
\mu_\ell \mid a_\ell, b_\ell &\overset{i.i.d}{\sim} \mathrm{Gamma}\,(a_\ell, b_\ell), \\
\alpha_{k,\ell} \mid e_{k,\ell}, f_{k,\ell} &\overset{i.i.d}{\sim} \mathrm{Gamma}\,(e_{k,\ell}, f_{k,\ell}), k, \ell = 1, \ldots, K.
\end{aligned}$$

For the computational implementation of the model, we developed a Markov chain Monte Carlo (MCMC) method and a stochastic gradient variational inference (SVI) algorithm, which updates the variational parameters based on minibatches of the dataset.

## 4 Key Results

The data is generated from a multivariate Hawkes process with $K = 2$ dimensions and the following parameter settings: $\boldsymbol{\alpha} = \begin{bmatrix} 0.6 & 0.15 \\ 0.3 & 0.6 \end{bmatrix}, \boldsymbol{\mu} = \begin{bmatrix} 0.05 \\ 0.1 \end{bmatrix}, T = 15000$. For the triggering kernels, we consider a scenario where the Beta mixture component is generated from a mixture of two Beta kernels:

$$\tilde{\phi}_{k,\ell}(t) = \varepsilon_{\text{true}} \exp(-t) + (1 - \varepsilon_{\text{true}}) \exp(-\lambda^{j,k} t),$$

where $\begin{bmatrix} \lambda^{11} & \lambda^{12} \\ \lambda^{21} & \lambda^{22} \end{bmatrix} = \begin{bmatrix} 2 & 0.8 \\ 0.8 & 2 \end{bmatrix}$. Additionally, $\varepsilon_{\text{true}} = \{0, 0.5, 1\}$ represents the true degree of information borrowing across dimensions. We fit the model using two methods: a Metropolis-within-Gibbs exact full-batch sampler based on Markov chain Monte Carlo (MCMC) and a stochastic gradient variational inference (SVI) algorithm based on minibatches. We also compare our method with random $\varepsilon$ ('RANDOM'), to two benchmark versions of both MCMC and SVI where there is no information borrowing ('IDIO') or the triggering kernels are identical ('COMMON'). Additionally, we also consider a frequentist benchmark method based on piecewise basis kernels using the EM algorithm (EM-BK, see Zhou et al., 2013). We evaluate both the point and uncertainty estimation accuracy for the triggering kernels, using a set of performance metrics. We use the root mean integrated sqaured error (RMISE) as a metric for point estimation accuracy:

$$\text{RMISE}(\phi) = \frac{1}{K^2} \sum_{k=1}^{K} \sum_{\ell=1}^{K} \sqrt{\int_0^{+\infty} \left( \phi_{k,\ell}^{\text{true}}(x) - \hat{\phi}_{k,\ell}(x) \right)^2 \, \mathrm{d}x}$$

Figure 1 shows the results and their specific scenarios. It can be shown that our methods have the lowest estimation error for both methods under most scenarios (except for MCMC when $\varepsilon = 1$, where the difference is very small). For the computation costs, SVI took only less than hour to converge while the MCMC algorithm takes over a day, further suggesting that the usage of scalable stochastic variational algorithm methods dramatically increases computationally efficiency.

| | MCMC | | | SVI | | |
| | RANDOM | IDIO | COMMON | RANDOM | IDIO | COMMON |
|---|---|---|---|---|---|---|
| $\varepsilon = 0$ | **0.060** | 0.063 | 0.254 | **0.159** | 0.170 | 0.289 |
| | (0.01) | (0.009) | (0.001) | (0.029) | (0.018) | (0.01) |
| $\varepsilon = 0.5$ | **0.055** | 0.061 | 0.130 | **0.152** | 0.160 | 0.162 |
| | (0.013) | (0.007) | (0.002) | (0.017) | (0.022) | (0.038) |
| $\varepsilon = 1$ | 0.027 | 0.061 | **0.023** | **0.087** | 0.105 | 0.097 |
| | (0.004) | (0.009) | (0.004) | (0.060) | (0.012) | (0.064) |

Table 1: RMISE as point estimation metric for all methods under true information-borrowing ratios. The values in the grid cells are averaged over 10 independently datasets, and the standard deviation is shown in the brackets. The numbers in bold refers to the best-performing method among 'RANDOM', 'IDIO' and 'COMMON' for both MCMC and SVI methods.

## 5 Conclusion

In this work, we developed a novel Multivariate Hawkes processes model for complex temporal event data. Especially, we flexibly modeled the decaying patterns of the triggering kernels using a Dependent Dirichlet process mixture of Beta distributions. Our model shows favorable results in both point and uncertainty prediction methods compared to benchmark models, and could serve as a basis for forecasting and decision making. We developed MCMC and SVI methods for computation, with SVI being computationally efficient and scalable to large datasets.

# A Kullback-Leibler property of the DDP mixture prior

We first prove the Kullback-Leibler property for a generic Dirichlet process mixture of Beta kernels model in Theorem 1, and then extend to our model setting in Corollary A.1.

**Theorem 1.** *Let $\mathcal{H}_0$ be the set of all continuous densities on $[0, 1]$. If $f_0(x) \in \mathcal{H}_0$, $\Pi$ is the prior induced by the Beta likelihood mixture kernel and $DP(\alpha_{DP}, \mathrm{Gamma}\,(c_a, d_a) \times \mathrm{Gamma}\,(c_b, d_b))$ on $\mathcal{H}_0$, Then $f_0 \in KL(\Pi)$, i.e. there exists $\varepsilon > 0$ such that $\Pi(\{g : KL(f_0, g)\}) < \varepsilon$.*

*Proof.* We show the KL property of $f_0$ by proving Conditions A1-A3 from Theorem 1 in Wu and Ghosal (2008) holds. Since we don't have $\phi$ in our case, Condition A2 is automatically satisfied. The remaining proof is similar to Theorem 11 in (Wu and Ghosal, 2008), except that the mixing distribution is drawn from a Dirichlet process, and that the two location parameters are also being mixed. For $\forall f_0 \in \mathcal{H}_0$ and $\forall \varepsilon > 0$, there exists a finite Beta mixture function $f_{P_\varepsilon}$ with $H$ mixtures where

$$f_{P_\varepsilon}(x) = \sum_{k=1}^{H} \frac{f_0\left(\frac{k-1}{H-1}\right)}{\sum_{k=1}^{H} f_0\left(\frac{k-1}{H-1}\right)} f_{\mathrm{Beta}}(x \mid k, H-k) = \int f_{\mathrm{Beta}}(x \mid a, b) dP_\varepsilon, \tag{4}$$

and

$$P_\varepsilon = \sum_{k=1}^{H} \omega_k \delta_{a,b}(a_k^0, b_k^0), \quad \omega_k^0 = \frac{f_0\left(\frac{k-1}{H-1}\right)}{\sum_{k=1}^{H} f_0\left(\frac{k-1}{H-1}\right)}, \quad a_k^0 = k, \quad b_k^0 = H-k, \tag{5}$$

such that

$$\int_0^1 f_0(x) \log \frac{f_0(x)}{f_{P_\varepsilon}(x)} dx < \varepsilon.$$

Thus Condition A1 holds. We then show Condition A3 also holds. First, we define $\mathcal{C}_\omega \subset \mathbb{S}^{H-1}, \mathcal{C}_{ab}^h \subset \mathbb{R}^2, h = 1, \ldots, H$ as sets such that

$$\mathcal{C}_\omega = \left\{ (\omega_1, \ldots, \omega_H) : \omega_h > \omega_h^0 e^{-\frac{\varepsilon}{4}}, \quad \sum_{h=1}^{H} \omega_h = 1 \right\},$$

$$\mathcal{C}_{ab}^h = \Big\{ (a_h, b_h) : a_h < a_h^0, b_h < b_h^0,$$
$$(a_h^0 - a_h)\left(\log x_M + \psi(a_h + b_h) - \psi(a_h)\right) + (b_h^0 - b_h)\left(\log(1 - x_M) + \psi(b_h + a_h) - \psi(b_h)\right) < \frac{\varepsilon}{4} \Big\}, \tag{6}$$

where

$$x_M = \max\left\{ d, 1 - d, \frac{a_h^0 - a_h}{a_h^0 - a_h + b_h^0 - b_h} \right\}.$$

Finally, we let $\mathcal{C}_{ab} = \oplus_{h=1}^{H} \mathcal{C}_{ab}^h$, $\boldsymbol{\omega} := (\omega_1, \ldots, \omega_H) \in \mathbb{S}^{H-1}$, $\boldsymbol{a} := (a_1, \ldots, a_H) \in \mathbb{R}^H$, $\boldsymbol{b} := (b_1, \ldots, b_H) \in \mathbb{R}^H$ and let $\mathcal{W} \subset \mathcal{H}$ be the set of finite mixture distributions induced by $\mathcal{C}_\omega$ and $\mathcal{C}_{ab}$:

$$\mathcal{W} := \left\{ P \in \mathcal{W} \mid P = \sum_{k=1}^{H} \omega_k \delta_{a,b}(a_k, b_k), \boldsymbol{\omega} \in \mathcal{C}_\omega, (a_h, b_h) \in \mathcal{C}_{ab}^h \right\}.$$

We note that for the chosen $\boldsymbol{\omega}_0, \boldsymbol{a}_0, \boldsymbol{b}_0$, for any $\omega \in \mathcal{C}_\omega$, we have

$$\frac{\sum_{j=1}^{H} \omega_h^0 f_{\mathrm{Beta}}\left(x \mid a_h^0, b_h^0\right)}{\sum_{j=1}^{H} \omega_h f_{\mathrm{Beta}}\left(x \mid a_h^0, b_h^0\right)} < e^{\frac{\varepsilon}{4}}, \quad \forall x \in (d, 1-d). \tag{7}$$

We then note that

$$(a_h^0 - a_h)\log x + (b_h^0 - b_h)\log(1 - x) \leq (a_h^0 - a_h)\log x_M + (b_h^0 - b_h)\log(1 - x_M), \quad \forall x \in (d, 1-d). \tag{8}$$

Thus for any $(a_h, b_h) \in \mathcal{C}_{ab}^h$, consider the Cauchy remainder form of the first-order expansion for $l(x, a_h^0, b_h^0) := \log f_{\text{Beta}}(x; a_h^0, b_h^0)$ at $(a_h, b_h)$, we have

$$
\begin{aligned}
l(x; a_h^0, b_h^0) &= l(x; a_h, b_h) + \nabla l(x; a_h, b_h)^T \begin{bmatrix} a_h^0 - a_h \\ b_h^0 - b_h \end{bmatrix} + \frac{\theta^2}{2} \begin{bmatrix} a_h^0 - a_h & b_h^0 - b_h \end{bmatrix} \nabla^2 l(x; a_h^*, b_h^*) \begin{bmatrix} a_h^0 - a_h \\ b_h^0 - b_h \end{bmatrix} \\
&< l(x; a_h, b_h) \\
&\quad + (a_h^0 - a_h)(\log x + \psi(a_h + b_h) - \psi(a_h)) + (b_h^0 - b_h)(\log(1 - x) + \psi(b_h + a_h) - \psi(b_h)) \\
&< l(x; a_h, b_h) + (a_h^0 - a_h)\log x_M + (b_h^0 - b_h)\log(1 - x_M) \\
&< l(x; a_h, b_h) + \frac{\varepsilon}{4}, \quad \forall x \in (d, 1 - d).
\end{aligned}
$$

(9)

Note that the chain of inequalities are based on equations (6) and (8), and the fact that

$$
\nabla^2 l(x; a, b) \prec 0, a > 0, b > 0.
$$

Finally, note that (9) is equivalent to

$$
\frac{f_{\text{Beta}}\left(x \mid a_h^0, b_h^0\right)}{f_{\text{Beta}}\left(x \mid a_h, b_h\right)} < e^{\frac{\varepsilon}{4}}, \quad \forall x \in (d, 1 - d),
$$

(10)

and if for $h = 1, \ldots, H$, $(a_h, b_h) \in \mathcal{C}_{ab}^h$, for any $(\omega_1, \ldots, \omega_H) \in \mathcal{C}_\omega$, we have

$$
\frac{\sum_{j=1}^H \omega_h f_{\text{Beta}}\left(x \mid a_h^0, b_h^0\right)}{\sum_{j=1}^H \omega_h f_{\text{Beta}}\left(x \mid a_h, b_h\right)} < e^{\frac{\varepsilon}{4}}, \quad \forall x \in (d, 1 - d).
$$

Combining equations (7) and (10), we have, for any $\boldsymbol{\omega} \in \mathcal{C}_\omega, (a_h, b_h) \in \mathcal{C}_{ab}^h, h = 1, \ldots, H$, we have

$$
\frac{\sum_{j=1}^H \omega_h^0 f_{\text{Beta}}\left(x \mid a_h^0, b_h^0\right)}{\sum_{j=1}^H \omega_h f_{\text{Beta}}\left(x \mid a_h, b_h\right)} = \frac{\sum_{j=1}^H \omega_h^0 f_{\text{Beta}}\left(x \mid a_h^0, b_h^0\right)}{\sum_{j=1}^H \omega_h f_{\text{Beta}}\left(x \mid a_h^0, b_h^0\right)} \cdot \frac{\sum_{j=1}^H \omega_h f_{\text{Beta}}\left(x \mid a_h^0, b_h^0\right)}{\sum_{j=1}^H \omega_h f_{\text{Beta}}\left(x \mid a_h, b_h\right)} = e^{\frac{\varepsilon}{2}},
$$

Thus

$$
\int_d^{1-d} f_0(x) \log \frac{f_{P_\varepsilon}(x)}{f_P(x)} dx < \int_d^{1-d} f_0(x) dx \cdot \frac{\varepsilon}{2} < \frac{\varepsilon}{2}.
$$

(11)

We then consider the scenario where $x \in (0, d] \bigcup [1 - d, 1)$. We want to show that the likelihood ratio $\frac{f_{\text{Beta}}\left(x \mid a_h^0, b_h^0\right)}{f_{\text{Beta}}\left(x \mid a_h, b_h\right)}$ has a uniform finite upper bound over all $x \in (0, d] \bigcup [1 - d, 1)$ and $(a_h, b_h) \in \mathcal{C}_{ab}^h, h = 1, \ldots, H$, i.e.

$$
\begin{aligned}
\sup_{x \in (0, d] \bigcup [1-d, 1)} \frac{f_{\text{Beta}}\left(x \mid a_h^0, b_h^0\right)}{f_{\text{Beta}}\left(x \mid a_h, b_h\right)} &= \frac{\text{Be}(a_h, b_h)}{\text{Be}(a_h^0, b_h^0)} \sup_{x \in (0, d] \bigcup [1-d, 1)} x^{a_h^0 - a_h}(1 - x)^{b_h^0 - b_h} \\
&= \frac{\text{Be}(a_h, b_h)}{\text{Be}(a_h^0, b_h^0)} d^{a_h^0 + b_h^0 - a_h - b_h} \leq \frac{\text{Be}(a_h, b_h)}{\text{Be}(a_h^0, b_h^0)},
\end{aligned}
$$

which follows from the fact that $a_h < a_h^0, b_h < b_h^0$. Thus we have

$$
\sup_{\substack{\boldsymbol{a}, \boldsymbol{b} \in \mathcal{C}_{ab} \\ \boldsymbol{\omega} \in \mathcal{C}_\omega}} \sup_{x \in (0, d] \bigcup [1-d, 1)} \frac{\sum_{j=1}^H \omega_h^0 f_{\text{Beta}}\left(x \mid a_h^0, b_h^0\right)}{\sum_{j=1}^H \omega_h f_{\text{Beta}}\left(x \mid a_h, b_h\right)} \leq e^{-\frac{\varepsilon}{4}} \sup_{\boldsymbol{a}, \boldsymbol{b} \in \mathcal{C}_{ab}} \frac{\text{Be}\left(a_h, b_h\right)}{\text{Be}\left(a_h^0, b_h^0\right)}
$$

$$
\leq \sup_{\boldsymbol{a}, \boldsymbol{b} \in \mathcal{C}_{ab}} \frac{\text{Be}\left(a_h, b_h\right)}{\text{Be}\left(a_h^0, b_h^0\right)} := M < +\infty.
$$

Thus we have

$$
\int_0^d f_0(x) \log \frac{f_{P_\varepsilon}(x)}{f_P(x)} dx + \int_{1-d}^1 f_0(x) \log \frac{f_{P_\varepsilon}(x)}{f_P(x)} dx < M\left(F_0(d) + 1 - F_0(1 - d)\right).
$$

Thus, we can choose $d$ small enough such that $M\left(F_0(d) + 1 - F_0(1 - d)\right) < \frac{\varepsilon}{2}$, such that

$$
\begin{aligned}
\int_0^1 f_0(x) \log \frac{f_{P_\varepsilon}(x)}{f_P(x)} dx &= \int_0^d f_0(x) \log \frac{f_{P_\varepsilon}(x)}{f_P(x)} dx + \int_{1-d}^1 f_0(x) \log \frac{f_{P_\varepsilon}(x)}{f_P(x)} dx + \int_d^{1-d} f_0(x) \\
&< \frac{\varepsilon}{2} + \frac{\varepsilon}{4} + \frac{\varepsilon}{4} = \varepsilon.
\end{aligned}
$$

Finally, as $\mathcal{C}_\omega, \mathcal{C}_{ab}$ are nonempty and open sets, we have $\Pi(\mathcal{W}) > 0$, thus $f_0 \in KL(\Pi)$. $\qquad \square$

**Corollary A.1.** *Consider $K$ continuous densities on $[0,1]$, i.e. let $f_0^1, \ldots, f_0^K \in \mathcal{H}_0$. Consider the following joint prior $\Pi^* : \oplus_{h=1}^H \mathcal{H}_0 \to \mathbb{R}$ for $(f^1, \ldots, f^K)$ such that*

$$f^k = \delta g_0 + (1-\delta)g^k, k = 1, \ldots, K,$$

$$g_k(x) = \int f_{\text{Beta}}(x \mid a, b)dP(a, b), k = 0, 1, \ldots, K,$$

$$P \sim \text{DP}(\alpha_{\text{DP}}, \text{Gamma}(c_a, d_a) \times \text{Gamma}(c_b, d_b)), \tag{12}$$

$$\delta \sim \text{Dirichlet}(1, 1),$$

*we have,*

$$\Pi^* \left( \{ f^1, \ldots, f^K : \text{KL}(f_0^k, f^k) < \varepsilon, k = 1, \ldots, K \} \right) > 0.$$

*Proof.* Let $f_0^0(x) := \frac{1}{K} \sum_{k=1}^K f_0^k(x)$. For $\varepsilon > 0$, we let

$$\mathcal{G}_0 = \{ g^0 \in \mathcal{H}_0 : \text{KL}(f_0^0, g^0) < \frac{\varepsilon}{2} \}, \quad \mathcal{G}^k = \{ g^k \in \mathcal{H}_0 : \text{KL}(f_0^k, g^k) < \frac{\varepsilon}{2} \}.$$

Note that from Theorem 1.1 we have $\Pi_{g^k} \left( g^k \in \mathcal{G}^k \right) > 0, k = 0, 1, \ldots, K$. Let $M := \max_{1 \le j, k \le M} \sup_{g^0 \in \mathcal{G}_0} \text{KL}(f_{ij}^0, g_0) < +\infty$, we have, based on the convexity of KL divergence:

$$\text{KL}(f_0^k, \delta g^0 + (1-\delta)g^k) \le \delta\text{KL}(f_0^k, g^0) + (1-\delta)\text{KL}(f_0^k, g^k) < \delta M + (1-\delta)\frac{\varepsilon}{2} < \varepsilon,$$

for all $\forall \delta < \frac{\varepsilon}{2M - \varepsilon}, g^k \in \mathcal{G}^k, k = 0, 1, \ldots, K$. Thus

$$\Pi^* \left( \{ f^1, \ldots, f^K : \text{KL} \left( f_0^k, f^k \right) < \varepsilon, k = 1, \ldots, K \} \right)$$
$$= \Pi^* \left( \{ g^0, g^1, \ldots, g^K, \delta : \text{KL} \left( f_0^k, \delta g^0 + (1-\delta)g^k \right) < \varepsilon, k = 1, \ldots, K \} \right)$$
$$> \Pi^* \left( \left\{ g^0, g^1, \ldots, g^K, \delta : g^k \in \mathcal{G}^k, k = 0, 1, \ldots, K, \delta < \frac{\varepsilon}{2M - \varepsilon} \right\} \right)$$
$$= \prod_{0 \le k \le K} \Pi_{g^k}(g^k \in \mathcal{G}^k)\Pi_\delta(\delta < \frac{\varepsilon}{2M - \varepsilon}) > 0.$$

$\square$

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
