# OpenReview forum: "MHP-DDP: Multivariate Hawkes Process with Dependent Dirichlet Process"
_NeurIPS.cc/2024/Workshop/BDU — NeurIPS BDU Workshop 2024 Poster_

### Official Review · Reviewer_K28p · 2024-09-26
**It's a short paper**

**Rating:** 5
**Confidence:** 5

**Review:**

The paper proposes a novel approach called MHP-DDP for modeling complex temporal dynamics among event sequences on multiple dimensions using Multivariate Hawkes Processes (MHPs). The proposed approach is based on a dependent Dirichlet process (DDP) and allows for flexible estimation of the excitation function by using a mixture of scaled Beta distributions. The paper presents two algorithms for implementing the MHP-DDP model and compares its performance to benchmark methods through simulations. The results show that MHP-DDP outperforms the benchmark methods in terms of lower estimation error, with the stochastic variational inference (SVI) algorithm being more computationally efficient than the Markov chain Monte Carlo (MCMC) algorithm.
Pros:
This paper presents a novel and adaptable model for Multivariate Hawkes Processes (MHPs) that enhances flexibility by allowing the sharing of information across dimensions and by estimating the excitation function using a mixture of scaled Beta distributions. This approach offers greater adaptability compared to traditional parametric models.
The study introduces two algorithms, Markov Chain Monte Carlo (MCMC) and Stochastic Variational Inference (SVI), for the implementation of the MHP-DDP model. Notably, the SVI algorithm proves to be computationally efficient and capable of scaling to accommodate large datasets.
The paper conducts a comprehensive evaluation of the proposed model, comparing it against established benchmark methods through detailed simulations. The findings highlight the MHP-DDP model's superior performance, particularly in achieving lower estimation errors.
Cons:
Please provide a deeper analysis of the limitations and potential drawbacks of the MHP-DDP model. What assumptions or conditions might impair the model’s performance?
Could you elaborate on the theoretical properties of the MHP-DDP model? Specifically, how does it compare to or differ from existing models cited in the literature?
To validate the MHP-DDP model's performance, please consider utilizing real-world applications, case studies, or datasets. Additionally, could you outline any plans for future research in this area?
I request more comprehensive details regarding the simulations used to compare the MHP-DDP model with benchmark methods. This should include the number of data points analyzed, the specific metrics applied for evaluation, and any underlying assumptions. Could you clarify which metrics were employed and explain their calculation methods?
Please provide additional information on the motivation behind the proposed approach and delineate how it diverges from existing methodologies.
Can you discuss the scalability of the proposed SVI algorithm with respect to data size and dimensional complexity? Have there been any tests of the algorithm on larger datasets?
Please offer further insights into the theoretical basis and advantages of employing a dependent Dirichlet process and a mixture of scaled Beta distributions within the MHP-DDP model.

---

### Official Review · Reviewer_LNaJ · 2024-10-04
**Blind Review**

**Rating:** 7
**Confidence:** 4

**Review:**

This paper proposes two algorithms to improve the efficiency of the MPH algorithm. The authors experimentally verify their effectiveness compared to the state of the art.

I am inclined to accept this paper. It covers a relevant topic area, the design seems novel compared to recent work, and the presentation is mostly clear. Performance is tested on previously used synthetic datasets, and the model is well-described.

A potential weakness of the paper is that the evaluation is limited to a single dataset. It would have been better to include higher-dimensional datasets, even if they were synthetic or from meme tracker datasets.

Other notes:

Line 120: There is a typo in “RMISE,” which stands for "Root Mean Integrated Squared Error (RMISE)."

---

### Decision · Program_Chairs · 2024-10-09

Accept (Poster)